# Evaluation of bioabsorbable calcium sulfate hemihydrate beads for local delivery of carboplatin

**Marine Traverson**[1]*, **Connor E. Stewart**[2], **Mark G. Papich**[3]

1 Department of Clinical Sciences, College of Veterinary Medicine, North Carolina State University, Raleigh, North Carolina, United States of America, 2 Department of Biological Sciences, College of Sciences, North Carolina State University, Raleigh, North Carolina, United States of America, 3 Department of Molecular Biomedical Sciences, College of Veterinary Medicine, North Carolina State University, Raleigh, North Carolina, United States of America

* matraver@ncsu.edu

**Data Availability Statement:** All relevant data are within the manuscript and its Supporting Information files.

**Funding:** The study was funded by North Carolina state University as part of Dr. Marine Traverson

## Abstract

The objectives of this study were to evaluate a novel kit of resorbable calcium sulfate beads marketed specifically for use in veterinary medicine and generally used for local delivery of antimicrobials as carboplatin-delivery system. The study characterized the elution of carboplatin *in vitro*, and investigated whether the initial dose and formulation of carboplatin, or the bead size significantly influences carboplatin elution *in vitro*. Calcium sulfate hemihydrate beads of 3- and 5-mm diameter were prepared. Five doses and two formulations of carboplatin (20, 50, 100, and 500 mg carboplatin per kit in powder formulation; 20 mg in liquid formulation) were tested in triplicates for each diameter beads. Beads were placed in 37˚C phosphate buffered saline for 72 hours. Carboplatin concentrations in the eluent were measured by high-performance liquid chromatography at 11 time points with a modified United States Pharmacopeia assay. Concentrations of carboplatin in the eluent proportionally increased with the initial dose and peaked between 13 and 52 hours, ranging from 42.1% to 79.3% of the incorporated load. Higher peak concentrations, percentages released, and elution rates were observed with the liquid formulation and with higher carboplatin doses. There was no significant difference in maximum carboplatin concentrations between 3-and 5-mm diameter beads, but 5-mm diameter beads had slower elution rates. The novel kit can be used for preparation of carboplatin-impregnated resorbable calcium sulfate beads at variable doses, sizes and formulations. Further study is warranted to define the *in vivo* requirements and effective carboplatin dose, spatial diffusion and desired duration of elution.

## Introduction

Carboplatin-impregnated biodegradable calcium sulfate beads are commercially available on the human pharmaceutical market (Osteoset resorbable mini bead kit, Wright Medical Technology Inc, Arlington, TN, USA; Stimulan Rapid Cure, Biocomposites Inc, Wilmington, NC,

start-up funds. The authors thank Kerrier for providing the calcium sulfate hemihydrate kits used in this study. The funders had no role in study design, data collection and analysis, decision to publish, or preparation of the manuscript.

**Competing interests:** The authors have no individual competing interests to declare. Kerrier provided the calcium sulfate hemihydrate kits used in this study. However, there are no patents, products in development or marketed products to declare. This does not alter our adherence to PLOS ONE policies on sharing data and materials.

USA) and compounded at an accredited veterinary compounding pharmacy (Wedgewood Pharmacy, Swedesboro, NJ, USA). These drug-delivery systems carry the advantage of allowing high and sustained release of carboplatin locally in the tissue, with minimal systemic exposure. These properties enable local tumor control after marginal and/or incomplete excision [1–9], which is not uncommon in veterinary medicine when the extent of the lesion and/or ethical and economic factors limit wide tumor excision. Their use has been described in previous veterinary studies, and *in vitro* elution of platinum from these products has been reported [1, 2, 10]. However, compounded drugs are typically not approved by the US Food and Drug Administration, and no routine safety or efficacy control is performed by the agency [4, 11]. A recent study showed that the amount of carboplatin found in 3-mm diameter calcium sulfate beads compounded at an accredited veterinary pharmacy (Wedgewood Pharmacy, Swedesboro, NJ, USA) was highly variable between lots with a mean content of 10.24 ± 1.84 mg carboplatin per bead, which was twice as high as the labeled strength of 4.6 mg/bead [4].

The amount of platinum delivered *in situ* has been variable between studies, with a peak delivery at 12 hours *in vitro* ranging from 27 to 89% of the incorporated carboplatin [1, 2, 10]. The size of the beads significantly influences the rates of dissolution and elution *in vitro*, with 4.8-mm diameter beads producing significantly higher dissolution rate and longer elution time than 3-mm diameter beads [12]. This difference in dissolution and elution rate between beads of different sizes has not been confirmed in regards to carboplatin delivery. However, higher peak percentage of incorporated carboplatin released was reported with 3-mm compared to 4.8-mm diameter beads [10].

A novel kit of resorbable high purity alpha hemihydrate calcium sulfate beads (Kerrier Local Antibiotic Delivery, Palm Beach Garden, FL, USA) has been marketed specifically for use in veterinary medicine and local delivery of antimicrobials. This novel kit offers the flexibility of preparing the beads on demand prior to use while mixing the commercialized calcium sulfate powder and mixing solution (sterile saline) with an appropriate antimicrobial drug at the desired dose. The paste obtained is then plated in 3.0- or 5.0-mm diameter bead molds until complete set, and is typically ready for use within 15 to 45 minutes depending on the antimicrobial and formulation selected. No data is available regarding the use of the kit for local delivery of platinum agents.

The objective of this study was to (1) evaluate the use of the novel kit generally use for the local delivery of antimicrobial agents as a carrier for local delivery of carboplatin, (2) characterize the elution of carboplatin from the novel carboplatin-impregnated calcium sulfate beads *in vitro*, (3) investigate whether the initial dose and formulation of carboplatin, or the size of the beads significantly influence the elution rate and percentage of total incorporated carboplatin released.

The effective *in vivo* tissue concentration of carboplatin carrying antitumor effects is unknown. *In vitro* studies have evaluated the half maximal inhibitory concentration ($IC_{50}$) of carboplatin for different neoplastic cell lines, which corresponds to the dose required to achieve 50% inhibition of neoplastic cell replication *in vitro* [13–17].

We hypothesized that the novel calcium sulfate hemihydrate kit would be suitable for delivery of carboplatin, and that the drug would elute over 3 days *in vitro* at concentrations greater than the $IC_{50}$ reported for different canine and feline neoplastic cell lines. We also hypothesized that the concentration of carboplatin in the eluent and the rate of elution would be positively associated with the dose and formulation of carboplatin as well as the size of the beads. While other studies have measured elemental platinum from elution experiments, we have chosen to measure intact carboplatin for this study as the release of platinum may be influenced by diverse factors that impact the degradation of carboplatin in the eluent.

## Materials and methods

### Pilot study

An initial pilot study was conducted to measure the standard elution profile of commercially available medical grade hemihydrate calcium sulfate beads (Absorbable bead kit, Kerrier Local Antibiotic Delivery, Palm Beach Garden, FL, USA; Ref 12–0125, Lot 04272016) impregnated with methylene blue (Thermo Fisher, Waltham, MA, USA). Methylene blue is often used in pilot elution experiments because it is readily available and can be quickly and easily measured with a spectrophotometer. Beads were prepared using a dose equivalent to 100 mg methylene blue per kit and allowed to elute in phosphate buffered saline (PBS) for six days. Temperature of 37˚C and constant agitation were maintained in a shaker incubator (New Brunswick C25 Incubator Shaker, Edison, NJ, USA). Samples were taken at 0, 1, 2, 3, and 6 days, and the concentration of methylene blue was calculated from the absorbance measured using a VersaMax microplate reader (Molecular Devices, San Jose, CA, USA) set at a wavelength of 663 nm and a standard curve created with serial dilutions of methylene blue. Peak elution concentration was achieved after approximately 3 days, which guided our study to a duration of 72 hours when using carboplatin-impregnated beads.

### Preparation of carboplatin-impregnated calcium sulfate hemihydrate beads

Five total formulations and doses of carboplatin were assessed: 20, 50, 100, and 500 mg carboplatin per kit in powder formulation (Sigma-Aldrich, St. Louis, MO, USA; Category C2538 Lot MKCK1243, C2538 MKCF9647, C1096407 R064PO) and 20 mg carboplatin per kit in liquid formulation (Accord Healthcare, Durham, NC, USA) at 10 mg/mL. Considering the absence of literature reference or guidelines describing the local concentration of carboplatin required in the tissues, the different doses tested were selected arbitrarily taking into account the amount of powder and liquid (carboplatin) that could safely be added to the calcium sulfate powder without compromising the production of the beads.

Each kit was divided in half before mixing to form 3.0- and 5.0-mm diameter beads using the manufacturer's standard preparation technique. Under a laminar flow biological safety cabinet (Thermo Fisher, Waltham, MA, USA), half of the total dose of carboplatin powder was added to half of the hemihydrate calcium sulfate powder, mixed with 2 mL mixing solution (half kit), and allowed to set for 60 min in the template until completely dry. The beads impregnated with the liquid formulation carboplatin were prepared using 1 mL of mixing solution and 1 mL of liquid formulation carboplatin (10 mg total). The procedure was repeated in triplicate for each bead size and carboplatin dose and formulation to obtain a mean of three measurements at each time point. Control beads of both sizes were prepared similarly, in triplicate, using a half kit of hemihydrate calcium sulfate powder with 2 mL mixing solution and no carboplatin. The beads were stored overnight in individual Erlenmeyer flasks at room temperature shielded from light. The following day, 200 mL PBS solution (pH 7.4) was added into each flask containing the beads, and the flask openings were covered with aluminum foil to prevent evaporative losses. Temperature and constant agitation were maintained at 37˚C and 50 revolution per minute (rpm) in a shaker incubator, according to the United States Pharmacopeia (USP) standards for dissolution testing [18]. The volume of the solvent used for dissolution was selected to avoid reaching the maximum thermodynamic solubility reported for carboplatin (15 mg/mL) [18].

### Samples collection

The elution profile of carboplatin was measured over 72 hours. Immediately before addition of PBS to each flask, 1 mL sample of PBS solution was collected (time 0). Subsequent samples

were collected at 1, 2, 3, 6, 9, 12, 24, 36, 48, and 72 hours. The samples were stored at -80°C until the end of the elution study. After each sample collection, 1 mL of fresh PBS was added to the flask to maintain a constant volume of 200 mL. The temperature and agitation of the incubator were recorded before each sample collection, and the volume of solution in each flask was assessed to correct for any variation. The beads' dissolution was assessed subjectively by visual inspection.

## Calculation of carboplatin concentration in the eluent

The concentration of carboplatin in each sample was evaluated using High Performance Liquid Chromatography (HPLC). The Agilent 1260 series HPLC system (Agilent Technologies, Santa Clara, CA, USA) consisting of a quaternary solvent delivery system, autosampler, and UV detector set at a wavelength of 220 nm was used for the analysis, and chromatograms were integrated in a computer program (Agilent OpenLab version C.01.09, Santa Clara, CA, USA). Retention times for the peaks of interest were measured between 3.6 and 3.7 minutes. Official USP analytic reference standard carboplatin (United States Pharmacopeia (USP), Rockville, Maryland, USA; Lot R01080, Catalog 1096407) was used to create a stock solution in 100% HPLC-grade distilled water at 1 mg/mL. Standard serial dilution series of the stock solution were generated with distilled water to prepare fortifying solutions for quality control and calibration. Eight calibration standards (range: 0 to 50 µg/mL) were completed by adding the fortifying solutions to the PBS solution. HPLC columns (Ace C18—AR stationary phase; Advanced Chromatography Technologies Ltd., Aberdeen, Scotland, UK) measuring 4.6 mm x 15 cm were used, and maintained at a constant temperature of 40°C. The mobile phase consisted of 100% HPLC grade distilled water (flow rate: 1 mL/min). A volume of 40 µL was loaded into HPLC vials for injection of all study samples, calibration samples, and control samples. The assay met acceptance criteria of the USP monograph for the testing of carboplatin [18]. The concentration obtained in a sample was multiplied by the total eluent volume (i.e. 200 mL) at the time of sample collection to calculate the total amount of carboplatin released in PBS. The result was expressed as a percentage of the initial dose of carboplatin incorporated into the beads.

## Statistical analysis

All statistical analyses were performed using SAS software (Version 9.4, SAS, Cary, NC, USA). Triplicate flasks were used for each carboplatin dose, formulation, and bead size, according to the solution protocols used in previous elution studies with a minimum statistical power of 80% [1–3]. This required results of the three replicates for each dose, formulation, and bead size to be reported as mean ± standard deviation. No outliers were removed from the analysis and there was no missing data.

Because the experiment was designed factorially with three replicates at each time point, a repeated-measures analysis of variance was used to analyze the percentage of carboplatin released. Fixed effects were all between-subject effects and included carboplatin dose, bead size, time, and all interactions between the three. A random replicate effect was used to account for individual variation in the construction and homogeneity of the beads (a within-subject effect). The covariance matrix was treated as unstructured, which was the model suggested using Akaike's Information Criterion.

The peak percentage of carboplatin released, the peaking times of those maximal values, the rate of carboplatin released (calculated as the peak percentage released divided by the hours at which the peak percentage occurred), and the final concentrations were compared across doses and bead sizes using a two-factor ANOVA. Due to increasing variances with increasing

means (heteroskedasticity), the rate values were natural-log transformed before the model was run.

The difference between liquid and powder percentages of carboplatin released were compared using a repeated-measures ANOVA allowing for fixed effects of carboplatin dose, bead size, time, and all interactions between the three, and a random replicate effect to account for individual variation in the construction and homogeneity of the beads. The covariance matrix was treated as first-order autoregressive moving average (ARMA(1,1)), which was the model suggested using Akaike's Information Criterion.

In all linear models, results were considered statistically significant when $p < 0.05$. Once an effect was found to be statistically significant, least squares means were compared using the Tukey correction for multiple comparisons.

## Results

### Bead preparation and study completion

All beads set and were ready for use within 45 to 60 minutes of preparation independently of the dose or the formulation of carboplatin used. The 5.0-mm diameter beads tended to require longer setting time than the 3.0-mm diameter beads. The beads demonstrated minor dissolution throughout the study period, with a subjectively roughened surface starting at 24 hours, and a mild progressive opacification of the PBS. No subjective difference was observed between flasks of different carboplatin doses, formulations or bead sizes.

All flasks maintained a volume of 200 mL of PBS through the entire study period, with the exception of two flasks in the 500 mg equivalent carboplatin dose group. One flask containing 3.0-mm diameter beads decreased to 195 mL PBS volume, and one flask of 5.0-mm diameter beads to 190 mL PBS. Those changes were most likely related to evaporative losses, despite the fact that each flask was covered with an aluminum foil. The carboplatin concentration values for these flasks were therefore adjusted at those particular time points to reflect the actual quantity of carboplatin available in 195 and 190 mL of PBS. In other words, the total quantity of carboplatin obtained in 1 mL of sampled PBS solution analyzed by HPLC was multiplied by 195, and 190 respectively instead of 200 in order to obtain the corrected total quantity of carboplatin in those 2 flasks at these specific time-points. The temperatures of the incubator at the time of sample collection ranged from 36.9 to 37.1°C, and the oscillation ranged from 50 to 51 rpm.

### Carboplatin elution

**Effect of carboplatin dose.** The peak concentration of carboplatin achieved in the eluent increased proportionally with the carboplatin dose loaded into the beads ($p = 0.001$), with mean peak percentage of incorporated carboplatin released ranging from 42.1 to 79.3% of the initial dose (Table 1). The model indicated that carboplatin dose significantly impacted peaking time ($p = 0.021$), with a significant difference between the 500 mg dose, which had a mean time to peak of 13.5 hours, and all the other doses. The other doses did not differ significantly from each other, with mean peaking time ranging from 44 to 52 hours (all with SE = 3.9). Carboplatin dose had a significant effect on elution rate ($p < 0.001$). The highest dose (500 mg) released carboplatin at a significantly higher rate than all the other doses ($p < 0.001$ for all pairwise comparisons), but the lower doses did not differ significantly from each other ($p > 0.240$ for all pairwise comparisons) (Fig 1).

**Effect of carboplatin formulation.** When comparing powder to liquid formulations, the only carboplatin dose tested was 20 mg as higher doses would have exceeded the recommended amount of solution used per kit (when maintaining half dilution with the mixing

**Table 1. Peak carboplatin concentrations achieved in PBS after elution from calcium sulfate beads impregnated with different carboplatin doses.**

| Carboplatin dose (mg/kit) | Peak carboplatin concentration (μg/mL; mean ± SD) | | |
|---|---|---|---|
| | (peak percentage of incorporated carboplatin released) | | |
| | 3-mm beads | 5-mm beads | Both sizes combined |
| | (n = 3) | (n = 3) | (n = 6) |
| 0 | 0.02 ± 0.05 | 0.0 ± 0.00 | 0.0 ± 0.03 |
| | (0%) | (0%) | (0%) |
| 20 | 27.67 ± 4.17 | 21.07 ± 6.30 | 24.4 ± 6.00 |
| | (55.3%) | (42.1%) | (48.7%) |
| 50 | 67.77 ± 15.00 | 72.27 ± 13.87 | 70.02 ± 13.15 |
| | (54.2%) | (57.8%) | (56.0%) |
| 100 | 157.20 ± 20.06 | 172.76 ± 24.55 | 164.98 ± 21.79 |
| | (62.9%) | (69.1%) | (66.0%) |
| 500 | 991.33 ± 16.01 | 844.00 ± 141.27 | 903.83 ± 113.31 |
| | (79.3%) | (67.5%) | (72.3%) |

All carboplatin doses indicated refer to the powder formulation. Dilution factor: 0.

solution). The 3.0- and 5.0-mm diameter beads released a peak of 55.3% and 42.1% of incorporated carboplatin from the powder formulation, respectively, and 76.7% and 82.7% of incorporated carboplatin from the liquid formulation, respectively (Table 2). Beads prepared with the powder formulation produced a continuous carboplatin release over up to 52 hours, while a drop was observed with the liquid formulation after peaking at 9 and 12 hours for the 3.0- and 5.0-mm diameter beads, respectively (Fig 2). When holding bead size constant, the liquid formulation was associated with a higher peak carboplatin concentration achieved in the eluent

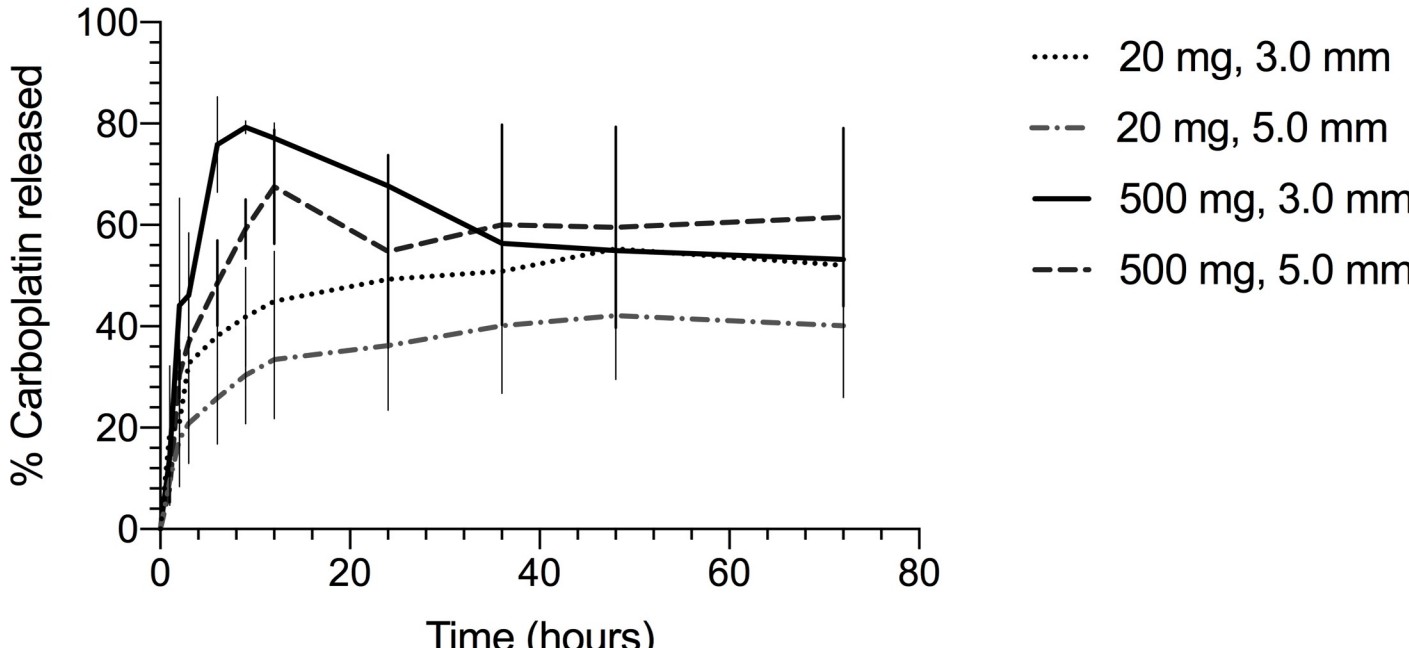

**Fig 1. Mean peak percentage of incorporated carboplatin released achieved in PBS after elution from calcium sulfate beads impregnated with different carboplatin doses.** Percentages are shown for 3-mm and 5-mm diameter beads. Powder formulation constant.

**Table 2. Mean peak carboplatin concentrations achieved in PBS after elution from calcium sulfate beads impregnated with different carboplatin formulations.**

| Carboplatin formulation | Peak carboplatin concentration (µg/mL; mean ± SD) (peak percentage of incorporated carboplatin released) | | |
| --- | --- | --- | --- |
| | 3-mm beads | 5-mm beads | Both sizes combined |
| | (n = 3) | (n = 3) | (n = 6) |
| Powder | 27.66 ± 4.17 | 21.07 ± 6.30 | 24.37 ± 6.00 |
| | (55.3%) | (42.1%) | (48.7%) |
| Liquid | 38.33 ± 3.51 | 41.33 ± 2.52 | 39.7 ± 2.73 |
| | (76.7%) | (82.7%) | (79.3%) |

Total dose of 20 mg carboplatin per kit maintained constant. Dilution factor: 0.

($p < 0.001$), a higher percentage of incorporated carboplatin released ($p < 0.001$), as well as a significantly higher elution rate ($p < 0.001$).

**Effect of bead size.** The model indicated that bead size did not significantly impact mean peak percentage of incorporated carboplatin released ($F_{(1,19)} = 0.72$, $p = 0.408$), nor mean percentage of incorporated carboplatin released at 72 hours ($F_{(1,19)} = 0.04$, $p = 0.708$). However, bead size had a significant impact on peaking time ($F_{(3,19)} = 21.37$, $p < 0.001$), without interaction with carboplatin dose ($F_{(3,16)} = 0.06$, $p = 0.980$). The mean time to peak for the 3-mm diameter beads was 35 hours (SE = 2.74), while the 5-mm diameter beads peaked at 44.8 hours (SE = 2.74). Bead size also significantly impacted elution rate ($F_{(1,19)} = 8.22$, $p = 0.010$), and smaller beads released carboplatin at a higher rate.

**Comparison to half maximal inhibitory concentrations of reported neoplastic cell lines.** Peak carboplatin concentrations achieved in the eluent with a half kit of calcium sulfate beads exceeded the reported *in vitro* carboplatin $IC_{50}$ values of various canine and feline neoplastic cell lines for most carboplatin doses. The reported $IC_{50}$ values of canine melanoma

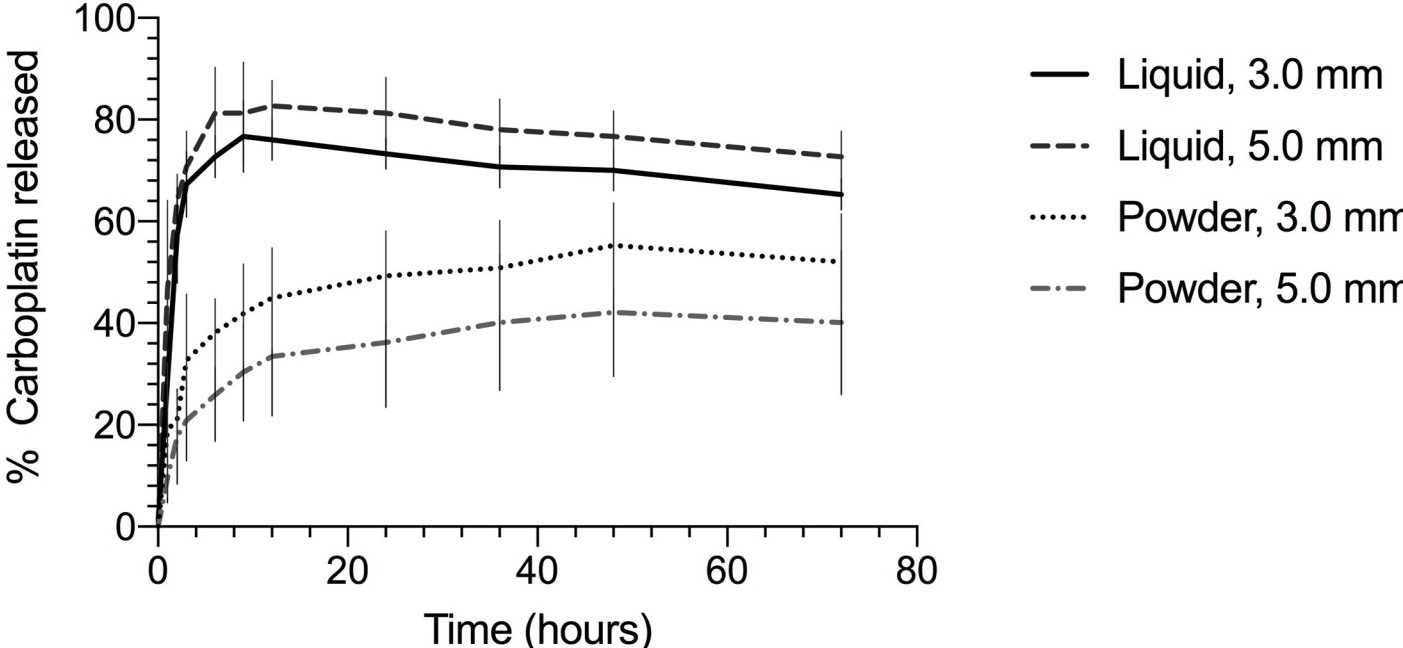

**Fig 2. Total percentage of incorporated carboplatin released achieved in PBS after elution from calcium sulfate beads impregnated with different carboplatin formulations.** Percentages are shown for 3-mm and 5-mm diameter beads. Carboplatin dose constant at 20 mg.

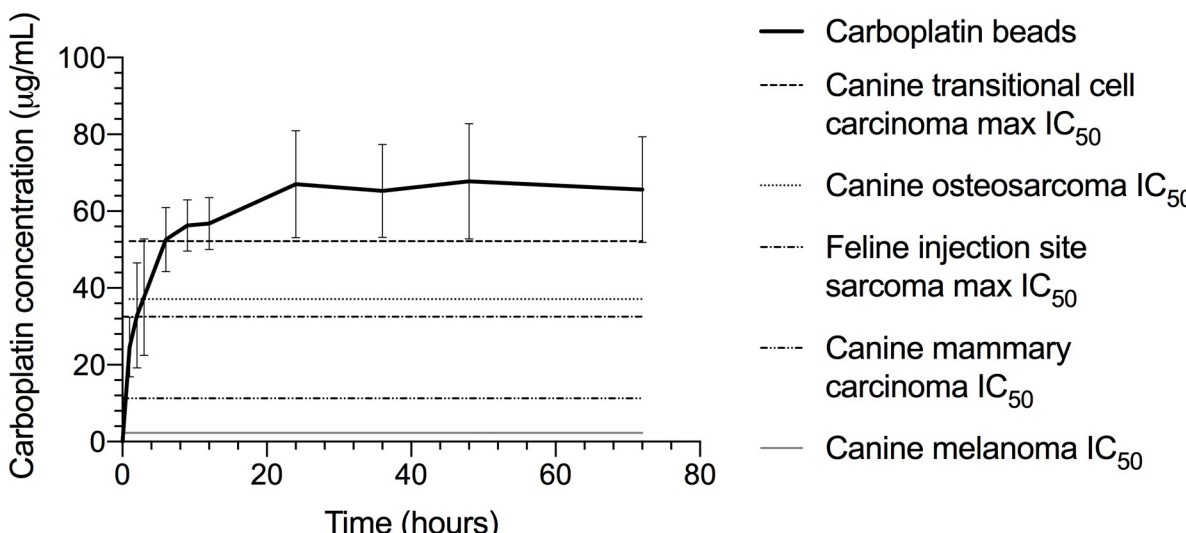

**Fig 3. Comparison between concentration of incorporated carboplatin released achieved in PBS after elution from calcium sulfate beads and half maximal inhibitory concentration (IC$_{50}$) of carboplatin for different neoplastic cell lines.** Carboplatin concentration is shown for 3-mm diameter beads, 50 mg carboplatin dose, and powder formulation.

(2.3 μg/mL; 6.1 μM) [15] and malignant mammary tumor (11.3 μg/mL; 30.5 μM) [13] were exceeded by all doses and carboplatin formulations at both bead sizes (Tables 1 and 2). The reported IC$_{50}$ values of feline injection site sarcoma (11.3–32.5 μg/mL; 30.5–87.5 μM) [16], canine osteosarcoma (37.1 μg/mL; 100 μM) [14], and canine transitional cell carcinoma (19.5–52.2 μg/mL; 52.4–140.5 μM) [17] were exceeded by the peak carboplatin concentrations obtained from the half kits mixed with 50, 100, and 500 mg equivalent carboplatin dose per kit (Fig 3).

## Discussion

This study evaluated the use of a novel kit of resorbable high purity alpha hemihydrate calcium sulfate beads (Kerrier Local Antibiotic Delivery, Palm Beach Garden, FL, USA) marketed specifically for use in veterinary medicine and generally used for local delivery of antimicrobials as a potential drug delivery system for carboplatin. The kit appeared suitable for use with carboplatin at variable doses and formulations, and required a total preparation time of 45 to 60 minutes. Carboplatin-impregnated calcium sulfate beads delivered peak *in vitro* carboplatin concentrations proportional to the initial carboplatin dose, with peak concentrations achieved predominantly at 48 hours and ranging from 42 to 79% of the initial incorporated carboplatin dose. Three main observations were made: (1) carboplatin concentrations in the eluent were significantly higher and the drug eluted faster when the liquid formulation was used as opposed to the powder formulation; (2) the bead size influenced the rate of carboplatin elution proportionally to the initial carboplatin dose but did not affect the total concentration of carboplatin eluted in solution; and (3) the percentage of total incorporated carboplatin released and the elution rate were proportionally higher as the initial carboplatin loading dose increased.

Carboplatin formulation had the largest effect on drug elution. Beads prepared with the liquid formulation released 1.4 to 2 times more carboplatin proportionally than beads prepared with the powder formulation. To our knowledge, no previous study described such a difference in dissolution or elution between liquid and powder formulations of any drug loaded

into calcium sulfate hemihydrate beads. The reasons for these differences are undetermined without further undue study. We ruled out pH differences as we used a buffered solution with a pH of 7.4. Both carboplatin formulations were added to the beads in a non-ionic form ($C_6H_{12}N_2O_4Pt$) without additional excipients. Yet, there was a difference in the proportion of mixing solution (sterile saline, NaCl 0.9%) and carboplatin solution ($C_6H_{12}N_2O_4Pt$ diluted in sterile water at 10 mg/mL), both having a pH of 5 to 7.

No statistical difference was observed in this study regarding the peak carboplatin concentration achieved in the eluent by the 3.0- and 5.0-mm diameter calcium sulfate beads, however 5.0-mm diameter beads showed a slower elution rate. The first result is discordant from another study [10], that demonstrated a higher percentage release from 4.8-mm compared to 3.0-mm diameter carboplatin-impregnated beads. Smaller beads (3.0-mm) did also release faster than larger beads (4.8-mm) when loaded with vancomycin [12]; however, one should use caution when comparing different drugs. Still, it is interesting to consider that different bead sizes may achieve different clinical goals, and small beads could be favored for rapid carboplatin release while large beads may be chosen if a prolonged delivery is required.

A large variation was observed in the carboplatin elution rate and pattern observed with 5.0-mm diameter calcium sulfate beads. This could be related to some degree of inconsistency in their preparation, and the fact that some of the 5.0-mm diameter molds were more commonly incompletely filled with calcium sulfate paste towards the edges of the plate when using a half kit volume, creating beads of inconsistent shapes and sizes. This was less of an issue when filling the 3.0-mm diameter molds. Each kit technically allows for the preparation of 113 beads of 5.0-mm diameter, or 333 beads of 3.0-mm. Half kits of calcium sulfate powder volume were used in this study. However, the total volume of the mixture varied according to the amount of carboplatin powder or solution added, and the mold plating resulted in variable numbers of beads per flasks. Beads were not individually counted as the total amount of carboplatin remained equivalent per groups of similar carboplatin dose and certain beads were not fully formed. However, disparity in plating may have influenced carboplatin elution results.

A significantly higher percentage of incorporated carboplatin released and faster carboplatin elution was observed as the initial carboplatin dose increased, with an earlier and steeper decay after 13.5 hours for the highest dose. To our knowledge, although predictable, no previous report raised such a correlation between high incorporated dose and high percentage release with carboplatin-impregnated calcium sulfate beads. This pattern has been observed with other drugs [19]. The high concentration gradients created by high carboplatin loads into each calcium sulfate bead likely caused a faster and higher carboplatin release [2, 20].

The peak percentage of incorporated carboplatin released was lower than previously demonstrated [1, 2], using the compounded veterinary product (Wedgewood Pharmacy, Swedesboro, NJ, USA). However, it remains above the mean percentages of 27 to 48% achieved after 96 hours in another study which evaluated products commercialized on the human market (Osteoset resorbable mini bead kit, Wright Medical Technology Inc, Arlington, TN, USA; Stimulan Rapid Cure, Biocomposites Inc, Wilmington, NC, USA) [10]. In this study, we used a high volume of beads compared to previous reports [1, 2]. The saturation of carboplatin in the environment certainly shouldn't have impacted the release as we followed the USP experimental guidelines for dissolution study to limit this risk. However, it is possible that the high volume of beads used in this study created an overcrowded environment that decreased the surface of contact with the eluent and thereby the elution of carboplatin.

We observed a decline in concentration of carboplatin over time after the maximum concentration was attained. We ruled out a dilution effect because the volume used to replace the aliquot of solvent withdrawn at each time point was insignificant and consistent with other study designs [10]. The decline is likely caused by degradation of the hydrophilic molecule in

solution. One of the strengths of this study is that we measured intact carboplatin in our analysis in opposition to other previously cited studies that measured elemental platinum. If carboplatin degraded in solution, presence of platinum would not reflect the true amount of intact carboplatin eluted.

A limitation of our study is that carboplatin elution was measured *in vitro* only and the *in vivo* release in a tissue environment may differ according to locoregional factors [2, 21, 22]. Therefore it is difficult to predict the effective *in vivo* carboplatin concentration. Indeed, the only published recommendation corresponds to the systemic maximum tolerated dose (MTD) established at 300 mg carboplatin/m$^2$ of body surface area in dogs (240 mg/m$^2$ in cats), leading to a peak plasma concentration of 200 μM equivalent to 80 mg carboplatin/L 4 to 6 hours after IV injection [23–25]. The desired local carboplatin dose may vary according to the microenvironment as well as the tumor type. *In vitro* studies have evaluated the susceptibility of diverse neoplastic cell lines to carboplatin [13–17]. Considering the IC$_{50}$ of common neoplastic cell lines, it appears that carboplatin concentrations achieved from the beads used in this study would inhibit all reported neoplastic cells. However, this is not a straight-forward extrapolation and the actual concentration in the tissues is influenced by the volume, the size of the beads, and the diffusion of carboplatin in the surrounding tissues. We recognize that the parallel drawn between carboplatin concentrations obtained in solution and the IC$_{50}$ of different neoplastic cell lines is highly artificial, and difficult to interpret clinically. However, we find it useful to know that the concentrations obtained in solution after elution of a half kit are in similar reference ranges than the reported IC$_{50}$ of classic neoplastic cell lines with which the kit could be used clinically. To our knowledge, no study has demonstrated a negative impact of a supra-MTD local dose of carboplatin. *In vivo* biocompatibility and safety studies are required to evaluate the systemic absorption of the drug, and rule out systemic toxicity.

The kit evaluated in this study has favorable advantages, including low expense, high versatility, wide availability, and biodegradable and biocompatible matrix. However, the powder formulation which has the potential to carry the highest carboplatin doses has to be purchased in a laboratory setting, restricting its use to compounding pharmacies. The liquid carboplatin formulation is the only accessible product to most veterinary clinics. Although it is proportionally associated with higher elution rate and concentration, the loaded dose is theoretically limited to 40 mg carboplatin per kit even if one were to maximize the carboplatin load and mix 4 mL of carboplatin solution total per kit (and discard the mixing solution) instead of 2 mL as we used in this study. Despite the lower carboplatin load obtained when using the liquid formulation, the resulting carboplatin concentration delivered may be sufficient to inhibit microscopic neoplastic cells *in vivo*, with low local and systemic toxicity.

Although common precautions should be used when handling chemotherapy drugs [26–28], it does not prevent the beads to be prepared on demand in the operating setting. But if local chemotherapy requirements in a hospital limit handling of the chemotherapy drugs to a biosafety cabinet and hood, it will limit the kit's use in the clinical setting. Such requirements may restrict its preparation to facilities equipped with a compounding pharmacy or biosafety assurances that meet the requirements of the local boards of pharmacy. Authorized pharmacies can prepare the beads under sterile conditions before being carried to the operating room. It is also possible to mix the beads with carboplatin beforehand and sterilize them prior to use by γ-irradiation or ethylene oxide [1, 29, 30].

## Conclusions

This commercially-available kit of resorbable high purity alpha hemihydrate calcium sulfate beads (Kerrier Local Antibiotic Delivery, Palm Beach Garden, FL, USA) marketed for use in

veterinary medicine and local delivery of antimicrobials could expand its use to include local tissue delivery of carboplatin. Doses ranging from 20 mg to 500 mg carboplatin per kit were tested, and would allow on demand preparation of carboplatin-impregnated calcium sulfate beads within 45 to 60 minutes of use with peak carboplatin delivery achieved within 48 hours in the local environment. The results reported here provide a foundation for future studies evaluating the *in vivo* efficacy of carboplatin treatment delivered with this product.

## Supporting information

**S1 Data.**
(XLSX)

**S1 Statistics.**
(PDF)

**S1 Statistics code.**
(DOCX)

## Acknowledgments

The authors thank Emily Griffith and James B. Robertson of the Office of Research at North Carolina State University for their expertise in performing the statistical analysis.

The authors thank Delta R. Dise, of the Clinical Pharmacology Laboratory at North Carolina State University for her expertise in performing the drug analysis, and Anne Crews, Laboratory Supervisor at the Adler Lab at North Carolina State University for her help with the elution study.

## Author Contributions

**Conceptualization:** Marine Traverson.

**Formal analysis:** Connor E. Stewart.

**Funding acquisition:** Marine Traverson.

**Investigation:** Marine Traverson, Connor E. Stewart, Mark G. Papich.

**Methodology:** Marine Traverson, Mark G. Papich.

**Project administration:** Marine Traverson.

**Resources:** Marine Traverson, Mark G. Papich.

**Supervision:** Marine Traverson.

**Validation:** Marine Traverson.

**Visualization:** Marine Traverson, Connor E. Stewart.

**Writing – original draft:** Marine Traverson, Connor E. Stewart.

**Writing – review & editing:** Mark G. Papich.

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
