## [Decision Letter · Decision Letter 0]

12 Aug 2020

PONE-D-20-21335

Evaluation of bioabsorbable calcium sulfate hemihydrate beads for local delivery of carboplatin

PLOS ONE

Dear Dr. Traverson,

Thank you for submitting your manuscript to PLOS ONE. After careful consideration, we feel that it has merit but does not fully meet PLOS ONE’s publication criteria as it currently stands. Therefore, we invite you to submit a revised version of the manuscript that addresses the points raised during the review process.

The manuscript was reviewed by three experts in the field, and they have all commended the article, but made some suggestions for minor modifications.

I therefore invite you to make the minor changes to the manuscript and resubmit it.

If you could write a response to reviewers that will aid in review upon resubmission.

I wish you the best of luck with your revisions.

Hope you are keeping safe and well in these difficult times.

We look forward to receiving your revised manuscript.

Kind regards,

Simon Clegg, PhD

Academic Editor

PLOS ONE

2. In the Methods section, please provide the product number and any lot numbers of the carboplatin purchased from Sigma-Aldrich and Accord Healthcare for your study.

3. In the Methods section, please provide the specific name, product number and any lot numbers of the hemihydrate calcium sulfate bead kit purchased from (Kerrier Local Antibiotic Delivery for your study.

4. To comply with PLOS ONE submission guidelines, in your Methods section, please provide additional information regarding your statistical analyses. For more information on PLOS ONE's expectations for statistical reporting, please see https://journals.plos.org/plosone/s/submission-guidelines.#loc-statistical-reporting.

5. Thank you for stating the following in the Financial Disclosure section:

"The study was funded by North Carolina state University as part of Dr. Marine Traverson start-up funds.

The authors thank Kerrier for providing the calcium sulfate hemihydrate kits used in this study.

We note that you received funding from a commercial source: Kerrier.

Reviewers' comments:

Reviewer's Responses to Questions

**Comments to the Author**

1. Is the manuscript technically sound, and do the data support the conclusions?

Reviewer #1: Yes

Reviewer #2: Yes

Reviewer #3: Yes

2. Has the statistical analysis been performed appropriately and rigorously? 

Reviewer #1: Yes

Reviewer #2: Yes

Reviewer #3: Yes

3. Have the authors made all data underlying the findings in their manuscript fully available?

Reviewer #1: Yes

Reviewer #2: Yes

Reviewer #3: No

4. Is the manuscript presented in an intelligible fashion and written in standard English?

Reviewer #1: Yes

Reviewer #2: Yes

Reviewer #3: Yes

5. Review Comments to the Author

Reviewer #1: very nice, highly interesting manuscript. Thank you for your work in this direction and field.

Minor thing: Page 8, line 156 the word "to" is missing

"The desired local carboplatin dose may vary according to the microenvironment as well as the tumor type." ==> In my opinion a graphic or at least a table showing desired doses as well as results to allow for an easier comparison would help.

You describe a loss of 5 ml or 10 ml PBS buffer solution - how is this accounted for? Do you see a specific reason?

Where do you describe the disadvantages of too high local concentrations or carboplatine?

Just an idea - have you read this: https://pubmed.ncbi.nlm.nih.gov/27067337/ ?

Reviewer #2: The authors have carried out the work on “Evaluation of bioabsorbable calcium sulfate hemihydrate beads for local delivery of carboplatin”

The work is mainly dealt on the in vitro elution of drug carboplatin. I have no specific concern on the article except in future experiment; the elution property of drug may be mimiced at in vivo model to justify in real situation. Reference 8 of the manuscript is to be checked.

Reviewer #3: Comments: Overall, the manuscript is well organized with a comprehensive presentation of carboplatin release from the beads. Below the are the comments.

General: For more clarity, a suggestion to revise the text, wherever applicable as follows “carboplatin, an anti-tumor agent is being investigated using these beads, which are generally used for the local delivery of anti-microbial agents”.

Introduction:

Line number: 45: What is the route of administration and what is the minimal systemic exposure. Please mention.

Materials and Methods:

Line number- 103: I understand the benefits of methylene blue, but why and how it can be compared with carboplatin release (3 days). Are they structurally relevant?

Line number 107: Please mention any literature justification for the selection of 4 doses for powder and 1 liquid formulation dose.

Line number 114: Please revise the text /typo “allowed to set for 60 min in the template”?

Line number 115: Please make it clear, 200ml pbs is added to the flask. Where are the beads, are they transferred to the flask?

Statistical analysis:

Line number 158: Delete the duplicate word “for each”

Line number 160- Grammar “was designed”

Line number 172- please align these parameters “fixed effects, dose, bead size and time” with line number 162

Results:

Line number 187- How many beads were there in each flask? And What could be the reason for volume reduction from 200ml to 190 and 195ml?

Line number 235- Carboplatin impregnated beads are placed in PBS and the eluted drug concentration is determined using HPLC which is compared with the IC50 values of carboplatin in different cell lines. It is not clearly written how this comparison is useful/hopeful for any further investigations (safety/cytotoxicity).

Discussion:

Line number 265 – “reason cannot be determined for the difference between powder and liquid carboplatin formulations behavior”. Any opinion why / which factors is influence this due to which further studies are required.

Table: What is the dilution factor?

6. PLOS authors have the option to publish the peer review history of their article (what does this mean?). If published, this will include your full peer review and any attached files.

Reviewer #2: No

Reviewer #3: No

---

## [Author Response · Author response to Decision Letter 0]

5 Oct 2020

Formatting of affiliation was modified (single space). We do not see any other style requirement mismatch; please indicate specifically as needed.

2. In the Methods section, please provide the product number and any lot numbers of the carboplatin purchased from Sigma-Aldrich and Accord Healthcare for your study.

Added lines 109-110 (USP). Unfortunately, the Carboplatin from Accord Healthcare was provided by the hospital pharmacy, and we do not have the product number.

3. In the Methods section, please provide the specific name, product number and any lot numbers of the hemihydrate calcium sulfate bead kit purchased from (Kerrier Local Antibiotic Delivery for your study.

Added line 93-94.

4. To comply with PLOS ONE submission guidelines, in your Methods section, please provide additional information regarding your statistical analyses. For more information on PLOS ONE's expectations for statistical reporting, please see https://journals.plos.org/plosone/s/submission-guidelines.#loc-statistical-reporting.

Statistical analysis software was reported line 164. Additional information regarding the statistical methods and results were reported lines 167-169, and 248-254.

5. Thank you for stating the following in the Financial Disclosure section:

"The study was funded by North Carolina state University as part of Dr. Marine Traverson start-up funds.

The authors thank Kerrier for providing the calcium sulfate hemihydrate kits used in this study.

We note that you received funding from a commercial source: Kerrier.

Modified accordingly.

Reviewers' comments:

Reviewer's Responses to Questions

Comments to the Author

1. Is the manuscript technically sound, and do the data support the conclusions?

Reviewer #1: Yes

Reviewer #2: Yes

Reviewer #3: Yes

2. Has the statistical analysis been performed appropriately and rigorously?

Reviewer #1: Yes

Reviewer #2: Yes

Reviewer #3: Yes

3. Have the authors made all data underlying the findings in their manuscript fully available?

Reviewer #1: Yes

Reviewer #2: Yes

Reviewer #3: No

4. Is the manuscript presented in an intelligible fashion and written in standard English?

Reviewer #1: Yes

Reviewer #2: Yes

Reviewer #3: Yes

5. Review Comments to the Author

Reviewer #1: very nice, highly interesting manuscript. Thank you for your work in this direction and field.

Minor thing: Page 8, line 156 the word "to" is missing

Line 165: Corrected.

"The desired local carboplatin dose may vary according to the microenvironment as well as the tumor type." ==> In my opinion a graphic or at least a table showing desired doses as well as results to allow for an easier comparison would help.

Thank you for your suggestion. A graphic comparing carboplatin concentration achieved in PBS after elution from the beads with IC50 of different neoplastic cell lines was added lines 266-269.

You describe a loss of 5 ml or 10 ml PBS buffer solution - how is this accounted for? Do you see a specific reason?

Changes were most likely related to evaporation, despite the fact that each flask was covered with an aluminum foil to prevent evaporative losses. The detail of the calculation has been added lines 202-208.

Where do you describe the disadvantages of too high local concentrations or carboplatin?

Just an idea - have you read this: https://pubmed.ncbi.nlm.nih.gov/27067337/ ?

To our knowledge, there is no description of the concept of “too high local concentrations”. As long as systemic absorption is limited, the use of supra-IC50 doses of carboplatin should not have any negative impact on the patient. Comment added lines 353-360.

Reviewer #2: The authors have carried out the work on “Evaluation of bioabsorbable calcium sulfate hemihydrate beads for local delivery of carboplatin”

The work is mainly dealt on the in vitro elution of drug carboplatin. I have no specific concern on the article except in future experiment; the elution property of drug may be mimicked at in vivo model to justify in real situation. Reference 8 of the manuscript is to be checked.

Thank you for this suggestion. Updated reference. Line 433-435.

Reviewer #3: Comments: Overall, the manuscript is well organized with a comprehensive presentation of carboplatin release from the beads. Below the are the comments.

General: For more clarity, a suggestion to revise the text, wherever applicable as follows “carboplatin, an anti-tumor agent is being investigated using these beads, which are generally used for the local delivery of anti-microbial agents”.

Modifications made in lines 20-21, 73-74, 273.

Introduction:

Line number: 45: What is the route of administration and what is the minimal systemic exposure. Please mention.

Thank you for your comment. The route of administration is implanted during surgery, as mentioned in the introduction. Unfortunately, we do not know what the systemic exposure is at this time because we have not implanted these beads into animals and measured the plasma concentrations. Therefore, we do not know the minimal systemic exposure; perhaps this could be done in future studies.

Materials and Methods:

Line number- 103: I understand the benefits of methylene blue, but why and how it can be compared with carboplatin release (3 days). Are they structurally relevant?

Thank you for your comment. Methylene blue is a very stable compound, and it is stable in solution. It cannot be directly compared to carboplatin, because they are different chemicals. However, we used it as a chemical marker because it is easy to measure as a dye. It provided preliminary evidence of the rate of release. 

Line number 107: Please mention any literature justification for the selection of 4 doses for powder and 1 liquid formulation dose.

To our knowledge, there is no particular literature reference or guidelines describing the local concentration of carboplatin required. Therefore, the dose was selected arbitrarily and taking into account the amount of powder and liquid that could safely be added to the calcium sulfate powder without compromising the production of the beads. Comment added Line 111-115.

Line number 114: Please revise the text /typo “allowed to set for 60 min in the template”?

Line 120: Corrected.

Line number 115: Please make it clear, 200ml pbs is added to the flask. Where are the beads, are they transferred to the flask?

Line 128: Comment added. The beads were stored in the flasks overnight, and PBS added into the flasks containing the beads at time 0.

Statistical analysis:

Line number 158: Delete the duplicate word “for each”

Deleted.

Line number 160- Grammar “was designed”

Line 170: Corrected.

Line number 172- please align these parameters “fixed effects, dose, bead size and time” with line number 162

Line 183-184: Adjusted.

Results:

Line number 187- How many beads were there in each flask? And What could be the reason for volume reduction from 200ml to 190 and 195ml?

Changes were most likely related to evaporation, despite the fact that each flask was covered with an aluminum foil to prevent evaporative losses. The detail of the calculation has been added lines 202-208.

Number of beads in each flask varied based on the plating technique and amount of carboplatin (powder or solution) added to the mixture. The fact that some of the beads were not fully formed and the amount of bead per flasks prevented counting. A comment was added in discussion (lines 309-315).

Line number 235- Carboplatin impregnated beads are placed in PBS and the eluted drug concentration is determined using HPLC which is compared with the IC50 values of carboplatin in different cell lines. It is not clearly written how this comparison is useful/hopeful for any further investigations (safety/cytotoxicity).

This parallel between carboplatin concentrations obtained in solution and the IC50 of different neoplastic cell lines is certainly highly artificial, and difficult to interpret clinically as the required local carboplatin dose is unknown and may vary according to multiple locoregional factors. However, we felt that it would be useful to ensure that the concentrations obtained in solution after elution of a half kit would at least find themselves in similar reference range than the reported IC50 of classic neoplastic cell lines with which the kit could be used clinically. Comment added line 353-360.

Discussion:

Line number 265 – “reason cannot be determined for the difference between powder and liquid carboplatin formulations behavior”. Any opinion why / which factors is influence this due to which further studies are required.

We are not sure how to explain these differences between the powder and the liquid formulation. It could be due to factors that are unknown. At this time, all we can say is that “the reason for these differences are undetermined without further undue study” (added lines 289-290). Such studies may include for example using the powder carboplatin and mixing it into the solution prior to preparation of the beads. This may not be as relevant as the carboplatin powder itself was mixed with sterile saline after addition to the calcium sulfate powder in our study. It is difficult to understand why such a difference was noted.

Table: What is the dilution factor?

 The dilution factor is 0 (no dilution). Added Lines 224 and 242.

---

## [Editor Report · Decision Letter 1]

20 Oct 2020

Evaluation of bioabsorbable calcium sulfate hemihydrate beads for local delivery of carboplatin

PONE-D-20-21335R1

Dear Dr. Traverson,

We’re pleased to inform you that your manuscript has been judged scientifically suitable for publication and will be formally accepted for publication once it meets all outstanding technical requirements.

Kind regards,

Simon Clegg, PhD

Academic Editor

PLOS ONE

Additional Editor Comments:

Many thanks for resubmitting your manuscript to PLOS One

As you have addressed all comments and the manuscript reads well, I have recommended it for publication

You should hear from the Editorial Office soon

It was a pleasure working with you and I wish you all the best for your future research

Hope you are keeping safe and well in these difficult times

Thanks

Simon

---

## [Editor Report · Acceptance letter]

27 Oct 2020

PONE-D-20-21335R1 

Evaluation of bioabsorbable calcium sulfate hemihydrate beads for local delivery of carboplatin 

Dear Dr. Traverson:

I'm pleased to inform you that your manuscript has been deemed suitable for publication in PLOS ONE. Congratulations! Your manuscript is now with our production department. 

Kind regards, 

on behalf of

Dr. Simon Clegg 

Academic Editor

PLOS ONE